# Rapid and Cost-Efficient Detection of *RET* Rearrangements in a Large Consecutive Series of Lung Carcinomas

**DOI:** 10.3390/ijms241310530

**Published:** 2023-06-23

**Authors:** Vladislav I. Tiurin, Elena V. Preobrazhenskaya, Natalia V. Mitiushkina, Aleksandr A. Romanko, Aleksandra A. Anuskina, Rimma S. Mulkidjan, Evgeniya S. Saitova, Elena A. Krivosheyeva, Elena D. Kharitonova, Mikhail P. Shevyakov, Ilya A. Tryakin, Svetlana N. Aleksakhina, Aigul R. Venina, Tatiana N. Sokolova, Aleksandr S. Martianov, Anna D. Shestakova, Alexandr O. Ivantsov, Aglaya G. Iyevleva, Evgeny N. Imyanitov

**Affiliations:** 1Department of Tumor Growth Biology, N.N. Petrov Institute of Oncology, 197758 St.-Petersburg, Russiaasokolova98@gmail.com (A.A.A.); shevmihail@icloud.com (M.P.S.); ilyatryakin@gmail.com (I.A.T.); anna.message19@gmail.com (A.D.S.); shurikiv@mail.ru (A.O.I.);; 2Department of Medical Genetics, St.-Petersburg Pediatric Medical University, 194100 St.-Petersburg, Russia

**Keywords:** NSCLC, fusion, tyrosine kinases, *RET*, PCR, NGS, thymidylate synthase, pemetrexed

## Abstract

*RET*-kinase-activating gene rearrangements occur in approximately 1–2% of non-small-cell lung carcinomas (NSCLCs). Their reliable detection requires next-generation sequencing (NGS), while conventional methods, such as immunohistochemistry (IHC), fluorescence in situ hybridization (FISH) or variant-specific PCR, have significant limitations. We developed an assay that compares the level of RNA transcripts corresponding to 5′- and 3′-end portions of the *RET* gene; this test relies on the fact that *RET* translocations result in the upregulation of the kinase domain of the gene and, therefore, the 5′/3′-end expression imbalance. The present study included 16,106 consecutive NSCLC patients, 14,449 (89.7%) of whom passed cDNA quality control. The 5′/3′-end unbalanced *RET* expression was observed in 184 (1.3%) tumors, 169 of which had a sufficient amount of material for the identification of translocation variants. Variant-specific PCR revealed *RET* rearrangements in 155/169 (91.7%) tumors. RNA quality was sufficient for RNA-based NGS in 10 cases, 8 of which carried exceptionally rare or novel (*HOOK1::RET* and *ZC3H7A::RET*) *RET* translocations. We also applied variant-specific PCR for eight common *RET* rearrangements in 4680 tumors, which emerged negative upon the 5′/3′-end unbalanced expression test; 33 (0.7%) of these NSCLCs showed *RET* fusion. While the combination of the analysis of 5′/3′-end *RET* expression imbalance and variant-specific PCR allowed identification of *RET* translocations in approximately 2% of consecutive NSCLCs, this estimate approached 120/2361 (5.1%) in *EGFR*/*KRAS*/*ALK*/*ROS1*/*BRAF*/*MET*-negative carcinomas. *RET*-rearranged tumors obtained from females, but not males, had a decreased level of expression of thymidylate synthase (*p* < 0.00001), which is a known predictive marker of the efficacy of pemetrexed. The results of our study provide a viable alternative for *RET* testing in facilities that do not have access to NGS due to cost or technical limitations.

## 1. Introduction

*RET* is an oncogene involved in the development of thyroid malignancies, non-small-cell lung carcinomas (NSCLCs) and several other categories of neoplasms [1]. *RET* activation may occur via gain-of-function missense mutations, which are highly characteristic for medullary thyroid carcinomas, or via gene translocation involving the kinase portion of this receptor. *RET* rearrangements occur in approximately 1–2% of adenocarcinomas of the lung. Recently approved RET-specific inhibitors, including pralsetinib and selpercatinib, render a clinical benefit for virtually all NSCLC patients with *RET* fusions [2,3,4]. In addition, instances of short-term responses of *RET*-rearranged NSCLCs have been documented for several multikinase inhibitors, whose spectrum of activity includes targeting RET kinase, e.g., vandetanib, cabozantinib, lenvatinib, sunitinib, alectinib, nintedanib, etc. [5,6,7,8]. *RET* testing may also guide the choice of cytotoxic therapy, given that *RET*-driven lung carcinomas demonstrate increased sensitivity to a pemetrexed-based treatment [7]. Consequently, *RET* analysis is a mandatory part of management of NSCLC patients [9].

The detection of *RET* rearrangements is complicated. Several studies revealed that immunohistochemical (IHC) staining for RET protein cannot be used even for preliminary evaluation of *RET* status, which makes a difference as compared to *ALK* and *ROS1* testing [4,6,10]. Furthermore, the break-apart fluorescence in situ hybridization (FISH) assay, which is a gold standard for the detection of *ALK*/*ROS1* rearrangements, has significant drawbacks when applied to the analysis of *RET* status [1,6,11,12,13,14]. Next-generation sequencing (NGS) is gradually replacing all other technologies for NSCLC molecular testing; however, it remains expensive, requires accumulation of multiple samples within the same run and is not yet easily available in many countries across the world [10,12]. Conventional PCR-based assays, being sensitive, specific and easily accessible, utilize variant-specific PCR and therefore cannot detect rare categories of *RET* rearrangements [1,4]. In addition, there are tests for 5′/3′-end unbalanced expression. Many tumor-driving chromosome rearrangements result in the overexpression of the kinase portion of the involved gene due to fusion with an actively transcribed gene partner; if this is the case, comparison of the amount of the 5′- and 3′-end RNA message of the involved gene will result in robust detection of actionable translocations [15]. Importantly, in contrast to the variant-specific PCR, this assay does not involve the analysis of a particular breakpoint or particular gene partner; therefore, in theory, it is capable of detecting all types of rearrangements.

Unlike IHC, FISH or NGS, the analysis of 5′/3′-end *RET* expression imbalance has not been subjected to comprehensive validation and is rarely utilized in clinical oncology [1]. This study aimed to develop a fast, efficient and non-expensive laboratory protocol that allows identification of both common and rare categories of *RET* oncogene fusions. Here, we present a pipeline which combines the advantages of the test for 5′/3′-end *RET* unbalanced expression and the variant-specific PCR for common *RET* gene rearrangements, and utilizes both these methods at the first step of the analysis. In addition to revealing NSCLCs with validated gene fusions, it allows the identification of ambiguous cases followed by clarification of their *RET* status with additional PCR assays and NGS. This robust approach led to the detection of a high number of *RET*-driven NSCLCs and allowed for describing the clinical characteristic of *RET*-associated tumors.

## 2. Results

cDNA of sufficient quality was obtained from 14,449/16,106 (89.7%) NSCLCs. The test for the 5′/3′-end unbalanced *RET* expression produced a deltaCt >/=3 in 184/14,449 (1.3%) analyzed cases. A sufficient amount of cDNA for further analysis was available in 169/184 (91.8%) NSCLCs with unbalanced *RET* expression. Testing for common *RET* rearrangements confirmed the presence of translocation in 152/169 (89.9%) NSCLCs *(KIF5B::RET (K15;R12): n* = 109; *CCDC6::RET (C1;R12): n* = 25;* KIF5B::RET (K16;R12): n* = 7;* KIF5B::RET (K22;R12): n* = 4;* KIF5B::RET (K23;R12): n* = 3;* NCOA4::RET (N8;R12): n* = 3;* KIF5B::RET (K24;R12): n* = 1). The analysis of the remaining 17 samples for 29 rare variants of *RET*-fusions identified 3 cases with translocations (*KIF13A::RET (K18-R12): n* = 1; *RELCH::RET (R10-R12): n* = 2). Two samples with 5′/3′-end *RET* unbalanced expression contained other actionable genetic events (*EGFR* mutation and *ALK* rearrangement, respectively) and, therefore, were not considered for NGS. Ten out of twelve remaining tumors had sufficient quality of material for NGS analysis. NGS identified *RET* rearrangements in eight of these cases, including six tumors with exceptionally rare fusion variants *(KIAA1217::RET (K13;R12): n* = 2; *CCSER2::RET (C9;R12): n* = 1; *RBPMS::RET (R6;R12): n* = 1;* RUFY3::RET (R11;R12): n* = 1;* NCOA4::RET (N9;R12): n* = 1) and two tumors with novel *RET* translocations (*HOOK1::RET (H21;R12)* and *ZC3H7A::RET (Z3;12)).*

Balanced 5′/3′-end *RET* expression was observed in 14,265/14,449 (98.7%) tumors. Eight common *RET* translocations were analyzed via variant-specific PCR in 4680 of these cases, and the presence of the rearrangement was detected in 33 (0.7%) NSCLCs *(KIF5B::RET (K15;R12): n* = 27; *CCDC6::RET (C1;R12): n* = 6). The analysis of these tumors revealed that the majority (*n* = 23) had a trend towards 5′/3′-end *RET* expression imbalance, although the deltaCt was below a predefined threshold of >/=3. Hence, four cases with a deltaCt = 2.8–2.9 for the 5′/3′-end *RET* expression ratio and that were negative for common *RET* fusions were tested for rare *RET* rearrangements using variant-specific PCR. This effort led to the identification of an NSCLC with a *KIAA1217::RET (K11;R12)* chimera. The distribution of fusion variants is described in Table 1 and Table 2. A flow-chart of the study is presented in Figure 1.

This study included 4849 tumors, which were analyzed both with a 5′/3′-end expression imbalance test and variant-specific PCR for eight common *RET* translocations (Table 3). These data provided an opportunity for a rough evaluation of the specificity and sensitivity of the former assay. It was revealed that the performance of the 5′/3′-end expression test was significantly influenced by the threshold chosen. For example, only 1.3% of the total number of analyzed carcinomas demonstrated 5′/3′-end imbalance with a deltaCt >/=3; 152/169 (89.9%) tumors meeting this cut-off carried common *RET* rearrangements; however, only 152/185 (82.2%) NSCLCs with *RET* fusions had a deltaCt satisfying this cut-off (Table 1 and Table 3). When we considered not only rearrangements detected via variant-specific PCR for eight common fusions, but also included *RET* alterations identified with other tests, the latter value did not change significantly and approached 163/197 (82.7%) (Table 1). When the threshold was brought down to >/=2, 4.6% of NSCLCs demonstrated expression imbalance, and the above estimates approached 170/224 (75.9%) and 170/185 (91.9%), respectively. The majority (176/185 (95.1%)) of common *RET* rearrangements were detectable if the 5′/3′ expression imbalance assay utilized the threshold >/=1; however, only 176/376 (46.8%) of tumors with this relaxed cut-off showed *RET* fusion (Table 3, Appendix A). 

We further analyzed the data for 3999 NSCLCs, which were comprehensively tested for all major driver alterations. Actionable genetic events were detected in 1758 (44.0%) tumors (*EGFR* mutations: 559 (14.0%); *KRAS* mutations: 751 (18.8%); *MET* exon 14 skipping: 60 (1.5%); *BRAF* mutations: 46 (1.1%); *ALK* fusions: 163 (4.1%); *ROS1* fusions: 58 (1.5%); *RET* fusions: 120 (3.0%)). *EGFR*/*KRAS*/*ALK*/*ROS1*/*BRAF*/*MET*-negative tumors accounted for 2361/3999 (59.0%) NSCLCs. Thus, *RET* rearrangements were detected in 120/2361 (5.1%) tumors negative for other common oncogenic alterations. There were no instances of the simultaneous presence of *RET* fusion and other actionable mutations. 

Clinical characteristics of patients with *RET* translocations were considered. The analysis of gender and smoking history revealed that *RET*-rearranged NSCLCs predominantly affected females and non-smokers; in this respect, the distribution of *RET*-driven NSCLCs resembled the ones for *ALK* and ROS translocations (Appendix A). Interestingly, while the median age of patients with *ALK*, *ROS1* and *RET* translocation was generally lower than in non-selected subjects with NSCLCs, this difference was less pronounced for *RET* than for *ALK* and *ROS1* (Appendix A).

Several clinical studies demonstrated increased sensitivity of *RET*-rearranged NSCLCs to pemetrexed [7,16,17]. This property can likely be attributed to reduced expression of thymidylate synthase (TYMS), which is a primary target for this drug [18,19]. Comparison of *TYMS* RNA expression in 88 *RET*-associated tumors and 234 NSCLCs lacking actionable genetic alterations revealed significantly lower expression of this gene in the former group (*p* < 0.00001, Figure 2A). Unexpectedly, this association was observed only for women (*p* < 0.00001, Figure 2B), while male patients showed similar *TYMS* expression in the *RET*-rearranged and wild-type NSCLCs (*p* = 0.276, Figure 2C). Furthermore, *RET*-associated NSCLCs showed decreased *TYMS* expression in females (*p* = 0.0385, Figure 2D), while there were no gender-related differences in the mutation-/fusion-negative tumors (*p* = 0.375, Figure 2E).

## 3. Discussion

Here, we present a non-expensive laboratory procedure, which consists of the combination of the test for the 5′/3′-end unbalanced *RET* expression and variant-specific PCR, and which relies on NGS only in instances of ambiguous results of PCR testing. In contrast to IHC or FISH, our methodology provides information on the identity of *RET* variants; therefore, by definition, this approach has a high level of specificity. However, it remains unclear whether some exceptionally rare *RET* translocations were missed in our data set. It is noteworthy that the test for the 5′/3′-end unbalanced *RET* expression per se has insufficient sensitivity, as approximately one out of six *RET* fusions were not detected by this test but were revealed by variant-specific assays. In this respect, the detection of *RET* fusions is similar to the analysis of *ROS1* rearrangements, where the comparison of 5′- and 3′-end expression has only modest sensitivity. In contrast, the 5′/3′-end unbalanced expression assay is highly reliable for the detection of *ALK* rearrangements [20]. In addition to the characteristics of expression imbalance, insufficient dissection of tumor cells may potentially compromise the results of this test. Overall, the observed frequency of *RET* rearrangements is on the upper limit of the interstudy variations; therefore, there is sufficient reason to believe that our approach is capable of detecting virtually all instances of *RET* fusions [12,21,22,23].

Systematic evaluation of the actual sensitivity of the proposed procedure requires its comparison with the “gold standard”, i.e., RNA-based NGS. It is particularly important to perform these validation experiments for NSCLCs, which have an increased probability of carrying a kinase-activating translocation, e.g., tumors diagnosed in young and/or female non-smokers that are negative for known driver mutations detected using standard methods [6,13,21,23,24]. By the time the manuscript was prepared, RNA-based NGS was accomplished for 87 NSCLCs, which satisfied the above clinical criteria and had no evidence for *ALK, ROS1, RET* or *NTRK1-3* gene rearrangements according to the 5′/3′-end expression test and variant-specific PCR. Strikingly, NGS analysis revealed that none of these carcinomas carried translocation in the above genes [24]. Despite these encouraging data, further efforts in avoiding even a minimal risk of false-negative results may be justified, especially given that the magnitude of benefit from targeted therapy in patients with *RET*-rearranged NSCLCs is indeed dramatic. This study utilized a combination of the 5′/3′-end unbalanced expression assay with a deltaCt >/=3 and one multiplexed PCR reaction for eight of the most common *RET* translocations as the screening procedure. It may be feasible to reduce the threshold for the deltaCt down to >/=2 or even to >/=1, as this will result only in a modest increase in the proportion of samples undergoing further analysis (Table 3). Comprehensive PCR-based testing of these samples is non-expensive, as it requires only three additional multiplexed reactions capable of recognizing 29 *RET* translocation variants. However, the relaxation of the deltaCt threshold will result in a dramatic increase in the number of samples, which were negative for 8 common and 29 rare translocations upon PCR analysis (Table 3, Appendix A). It is questionable whether PCR-negative samples with a deltaCt threshold above 1 but below 3 deserve to be subjected to NGS. The choice of the optimal cut-offs and the depths of the analysis of NSCLCs with borderline deltaCt values may depend on the local experience and the available resources.

This study failed to obtain cDNA of sufficient quality in 1657/16,106 (10.3%) NSCLCs. It is essential to emphasize that NSCLC specimens were collected from various hospitals; therefore, some inconsistencies in the processing and formalin fixation of tumor samples could have contributed to this relatively high failure rate. Surprisingly, most of the published large-scale studies did not report the proportion of samples unsuitable for *RET* analysis [10,13,14,18,22,23,25,26,27]. The available data indicate that the quality of NSCLC tissues is essential for all *RET* testing procedures. Tsuta et al. [21] considered 1927 samples obtained in a single cancer center and observed a FISH failure in 53 (2.8%) tumors. Similarly, only 2/121 (1.7%) were unsuitable for FISH testing in the study by Baker et al. [11]. Failure rates are apparently higher for RNA-based techniques. Reguart et al. [28] reported insufficient cDNA quality in 7/108 (6.5%) samples analyzed. The proportion of NSCLCs with degraded RNA approached 6/30 (20%) in the study by Piton et al. [29]. To our knowledge, there have been no investigations that rigorously compare failure rates across various *RET* testing platforms.

The analysis of *EGFR*/*KRAS*/*ALK*/*ROS1*/*BRAF*/*MET*-negative tumors revealed *RET* rearrangements in 5.1% tumors. This is a somewhat lower frequency as compared to that found in a recent large-scale study from China, which detected *RET* fusions in 8.9% NSCLCs with no other actionable alterations [13]. Drastic differences in the frequency of *EGFR* mutations between European and Asian patients may be a cause of these variations. Indeed, approximately half of Asian patients with NSCLCs carry *EGFR* mutations, so only a small proportion of Asian patients belong to the *EGFR*/*KRAS*/*ALK*/*ROS1*/*BRAF*/*MET*-wild-type subgroup [9,13]. Only 14.0% patients in our study carried *EGFR* mutations, and as many as 59.0% patients were negative for the above alterations. Consequently, while the frequency of *RET* rearrangements in non-selected NSCLCs is similar or somewhat higher than in Asian data sets [13,21,22,23], their presence in the *EGFR*/*KRAS*/*ALK*/*ROS1*/*BRAF*/*MET*-negative subgroup is less pronounced.

In contrast to several other reports, which observed rare instances of simultaneous occurrence of *RET* translocations and other actionable mutations [6,13], we did not detect tumors with the combination of *RET* fusion and alteration in *EGFR*, *KRAS*, *ALK*, *ROS1*, *BRAF* or *MET* oncogenes. Our data confirm that activating genetic events in the MAPK pathway are mutually exclusive in the vast majority of tumors, and these data may be taken into account while considering practical aspects of NSCLC testing [6,9]. The observed overrepresentation of young subjects, females and non-smokers among patients with *RET*-associated NSCLC is in good agreement with other reports [6,13,21,23,25]. However, similarly to other large studies, we have identified a significant number of subjects with *RET*-rearranged NSCLCs that do not meet the above criteria; therefore, all NSCLC cases should undergo comprehensive molecular testing irrespective of the clinical presentation of the disease [13,27,30].

Data on the reduced *TYMS* expression in *RET*-rearranged NSCLCs have been previously published by Song et al. [18]; however, the above report included only 11 tumors with *RET* translocation. The large size of our study permitted us to perform a more detailed analysis of relationships between *TYMS* status and the presence of *RET* fusion. Striking gender-related relationships were observed: low RNA expression of the *TYMS* gene was observed only in women with *RET*-associated NSCLCs, but not in men (Figure 2). This observation deserves subsequent studies in order to reveal whether some biologically active compounds, e.g., sex hormones or their antagonists, may potentiate the efficacy of pemetrexed.

The results of our study provide a viable alternative for *RET* testing for those facilities that do not have access to NGS due to cost or technical limitations. The proposed pipeline requires validation in independent laboratories.

## 4. Materials and Methods

This study included 16106 consecutive NSCLC patients, who were forwarded to the N.N. Petrov of Oncology (St. Petersburg, Russia) for molecular genetic testing between March, 2017 and February, 2023. The median age of included patients was 63 years, ranging from 17 to 97 years. There was 5396 (33.5%) females and 10,710 (66.5%) males. Areas enriched by the tumor cells were dissected from formalin-fixed paraffin-embedded tissues and subjected to DNA and RNA extraction [20]. Briefly, lysis of tumor sections was achieved via overnight incubation in 10 mM Tris–HCl (pH 8.0), 0.1 mM EDTA (pH 8.0), 2% SDS and 20mkl proteinase K (20 mg/mL) at 65 °C. Organic extraction involved shaking with 200 μL of Trizol and 90 μL of a chloroform–isoamyl alcohol mix (24:1), followed by centrifugation (15,000 g) for 15 min at 0 °C. The supernatant was transferred to a new tube. Nucleic acids were precipitated with 1 μL of glycogen (20 mg/mL) and 1 volume (300 μL) of cold isopropanol overnight at −20 °C, and then pelleted via centrifugation at 15,000 g for 30 min. Isopropanol was removed, and the precipitate was rinsed with 70% ethanol for 10 min. After thorough removal of ethanol, the precipitate was dried at 50 °C, and then dissolved in 10 μL of sterile water at 50 °C. cDNA was obtained through a reverse transcription reaction, which contained 5X reverse transcriptase reaction buffer, 200 units of RevertAid Reverse Transcriptase (Thermo Fisher Scientific Baltics UAB, Vilnius, Lithuania), 20 units of RiboCare RNase Inhibitor (Evrogen, Moscow, Russia), dNTP mix (20 nM each), random hexamers (0.25 μmol) and gene-specific primers for *RET* receptor tyrosine kinase. The mixture of RNA, dNTPs and primers was incubated for 5 min at 70 °C, 65 °C and 60 °C to achieve primer annealing, and then cooled at 0 °C for 2 min. To synthesize cDNA, the enzymes were added, and the reaction mix was incubated at 20 °C for 5 min, 38 °C for 30 min and 95 °C for 5 min. Primer sequences are described in Appendix A. The quality of cDNA was controlled through PCR amplification of the *SDHA*-specific transcript; samples with a cycle threshold (Ct) above 35 were considered unreliable for further analysis. All samples with the satisfactory results of the quality control were subjected to the test for 5′/3′-end unbalanced expression (Appendix A).

NSCLCs with a deltaCt >/= 3 were analyzed for 8 common RET translocation variants (*KIF5B::RET (K15;R12), CCDC6::RET (C1;R12), KIF5B::RET (K16;R12), KIF5B::RET (K22;R12), KIF5B::RET (K23;R12), KIF5B::RET (K24;R12), NCOA4::RET (N8;R12), NCOA4::RET (N7;R12)*) [31] using one multiplexed PCR reaction, 29 rare *RET* translocation variants (*CUX1::RET; FYCO1::RET; ITGA8::RET; KIF5B::RET; MPRIP::RET; RELCH::RET; SLC39A8::RET; TRIM33::RET; ZBTB41::RET; ADD3::RET; ANKS1B::RET; CCDC186::RET; FRMD4A::RET; KIAA1217::RET; KIF13A::RET; MYO5C::RET; RASSF4::RET; TBC1D32::RET; WAC::RET; CCNYL2::RET; LSM14A::RET; PCM1::RET; PRKG1::RET; PTPRK::RET; RUFY2::RET; SIRT1::RET; SLC25A36::RET; SORBS1::RET; TSSK4::RET*) using 3 multiplexed PCR reactions, and, wherever appropriate, NGS. The composition of multiplex PCR assays, primer and probes, and conditions for the PCR tests are described in Appendix A. In addition, 4680 NSCLC samples, which were received between March, 2022 and February, 2023, were tested for the 8 common gene rearrangements described above irrespective of the results of the 5′/3′-end unbalanced expression test. 

NGS RNA sequencing was performed using the custom QIAseq Targeted RNAscan Panel (Qiagen, Hilden, Germany), which is capable of detecting rearrangements in 6 genes (*ALK, ROS1, RET, NTRK1, NTRK2, NTRK3*). NGS was carried out on the Illumina MiSeq instrument. Fusions were detected using the Illumina RNA-Sequencing Alignment software (V.2.0.0) and STAR-Fusion pipeline (V.1.4.0).

Data on the status of *EGFR*, *KRAS*, *ALK*, *ROS1*, *BRAF* and *MET* oncogenes were available for 3999 NSCLCs. *EGFR* analysis was performed as described in [32]. The procedure for *ALK* and *ROS1* testing is given in detail in [20]. The detection of *MET* exon 14 skipping mutations was carried out in accordance with the study [33]. *KRAS* and *BRAF* mutation analysis is outlined in [34]. 

*TYMS* expression was measured using real-time PCR. Primer, probes and conditions for the PCR tests are described in Appendix A.

## Figures and Tables

**Figure 1 ijms-24-10530-f001:**
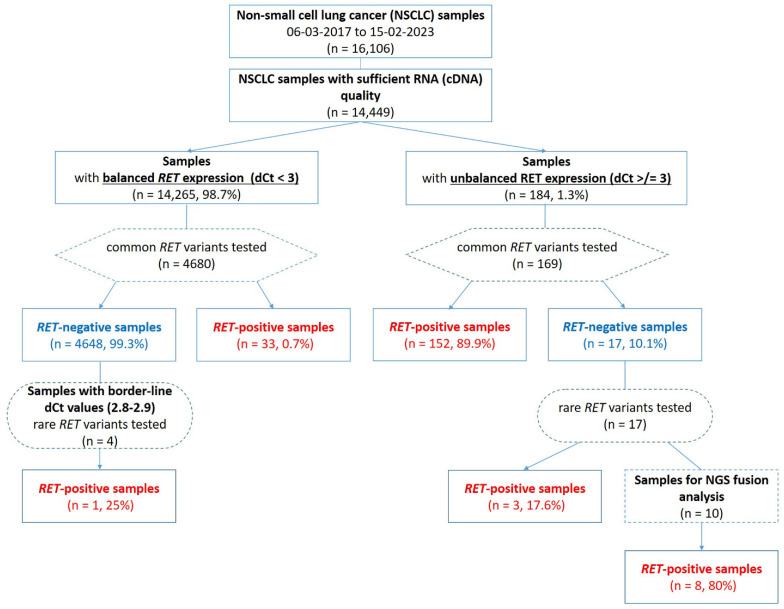
Workflow for detection of *RET* rearrangements in a large series of NSCLCs.

**Figure 2 ijms-24-10530-f002:**
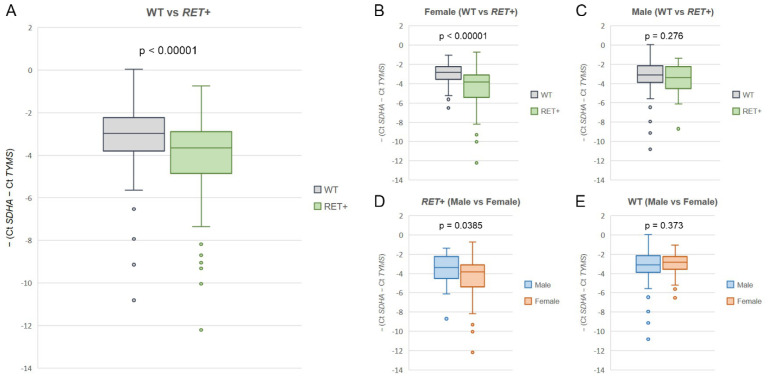
*TYMS* RNA expression in RET-rearranged and wild-type NSCLCs. (**A**) The analysis of the entire cohort of tumors with (*n* = 88) and without (*n* = 234) *RET* fusions; (**B**,**C**) comparison of *TYMS* RNA expression in females (wild-type: *n* = 73; *RET*-associated: *n* = 60) and males (wild-type: *n* = 161; *RET*-associated: *n* = 28); (**D**,**E**) gender-related differences in *TYMS* expression in *RET*-rearranged NSCLCs (males: *n* = 28; females: *n* = 60) and tumors lacking common driver mutational events (males: *n* = 161; females: 73). *p*-values were calculated using the Mann–Whitney *U* test.

**Table 1 ijms-24-10530-t001:** Distribution of *RET*-fusion-positive cases in samples with unbalanced and balanced *RET* 5′/3′-end expression.

	Cases with Unbalanced 5′/3′-End Expression of *RET*	Cases with Balanced 5′/3′-End Expression of *RET*	Total
Fusions detected via PCR for common variants	152 (82.2%)	33 (17.8%)	185
Fusions detected via PCR for rare variants	3 (75.0%)	1 (25.0%)	4
Fusions detected via NGS	8 (100%) (6–rare, 2–novel)	-	8
Total *RET*-fusion-positive cases (*n* = 197)	163 (82.7%)	34 (17.3%)	197

**Table 2 ijms-24-10530-t002:** Spectrum of identified RET fusions.

Fusion	Number of Cases
*KIF5B::RET (K15;R12)*	136 (69.1%)
*CCDC6::RET (C1;R12)*	30 (15.1%)
*KIF5B::RET (K16;R12)*	8 (4.1%)
*KIF5B::RET (K22;R12)*	4 (2.0%)
*NCOA4::RET (N8;R12)*	3 (1.5%)
*KIF5B::RET (K23;R12)*	3 (1.5%)
*RELCH::RET (R10;R12)*	2 (1.0%)
*KIAA1217::RET (K13;R12)*	2 (1.0%)
*KIF5B::RET (K24;R12)*	1 (0.5%)
*KIF13A::RET (K18;R12)*	1 (0.5%)
*KIAA1217::RET (K11;R12)*	1 (0.5%)
*CCSER2::RET (C9;R12)*	1 (0.5%)
*RBPMS::RET (R6;R12)*	1 (0.5%)
*RUFY3::RET (R11;R12)*	1 (0.5%)
*NCOA4::RET (N9;R12)*	1 (0.5%)
*ZC3H7A::RET (Z3;R12)*	1 (0.5%)
*HOOK1::RET (H21;R12)*	1 (0.5%)
Total	197

**Table 3 ijms-24-10530-t003:** The proportion of RET-rearranged tumors among NSCLCs with different dCt values as assessed using the 5′/3′-end expression imbalance test.

	dCt >/= 3	dCt >/= 2	dCt >/= 1	dCT < 1	Total
Cases tested for common variants	169	224	376	4473	4849
RET-positive (8 common variants)	152/169 (89.9%)	170/224 (75.9%)	176/376 (46.8%)	9/4473 (0.2%)	185/4849 (3.8%)

## Data Availability

The data that support the findings of this study are available from the corresponding author upon reasonable request.

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
