# Peer review of "Rapid and Cost-Efficient Detection of RET Rearrangements in a Large Consecutive Series of Lung Carcinomas"

_ijms, 2023, doi:10.3390/ijms241310530_

Round 1
Reviewer 1 Report
In the manuscript entitled "Rapid and cost-efficient detection of RET rearrangements in a 2 large consecutive series of lung carcinomas" by Vladislav I. Tiurin et al., the authors present the development of a 5’/3’-end expression imbalance assay for the RET gene, which was applied to a vast cohort of non-small cell lung cancer (NSCLC) cases. The study is well designed and the findings are in agreement with previously published RET-fusion prevalence in NSCLC. However, it would be desirable that the authors address some revisions in order to consider publication:
1- The authors used a predefined cutoff deltaCt value of ≥ 3, however, they do not provide the rationale for that cutoff value, nor do they provide a reference that can justify that threshold. A delta Ct of 3, corresponds to an 8-fold change difference between the 5’ and 3’ portion of the transcript, assuming 100% efficiency in PCR amplification. It would be desirable that the authors show the distribution of deltaCt values between confirmed RET-fusion negative cases and confirmed RET-fusion positive cases, to better define or confirm such deltaCt threshold. That data could be shown in the supplementary information
2- The authors should comply with the HUGO Gene Nomenclature Committee (HGNC) recommendations for the designation of gene fusions (PMID: 34615987). Thus, due to shortcomings of using in hyphen (-) or a forward slash (/) to describe gene fusions, HGNC has worked with the scientific community to determine a new, instantly recognizable and unique separator-a double colon (::) to be used in the description of fusion genes, and advocates its usage in all databases and articles describing gene fusions.
Author Response
Comment: The authors used a predefined cutoff deltaCt value of ≥ 3, however, they do not provide the rationale for that cutoff value, nor do they provide a reference that can justify that threshold. A delta Ct of 3, corresponds to an 8-fold change difference between the 5’ and 3’ portion of the transcript, assuming 100% efficiency in PCR amplification. It would be desirable that the authors show the distribution of deltaCt values between confirmed RET-fusion negative cases and confirmed RET-fusion positive cases, to better define or confirm such deltaCt threshold. That data could be shown in the supplementary information.
Response: We now comment on this issue in Results and Discussion sections, with the mention of newly added Table S3, Table S4 and Figure S1:
This study included 4849 tumors, which were analyzed both by 5’/3’-end expression imbalance test and by variant-specific PCR for eight common RET translocations. These data provided an opportunity for rough evaluation of the specificity and sensitivity of the former assay. It was revealed that the performance of 5’/3’-end expression test was significantly influenced by the threshold chosen. For example, only 1.3% of the total number of the analyzed carcinomas demonstrated the 5’/3’-end imbalance with deltaCt >/=3; 152/169 (89.9%) tumors meeting this cut-off carried common RET rearrangements, however only 152/185 (82.2%) NSCLCs with RET fusions had deltaCt satisfying this cut-off. When the threshold was brought down to >/=2, 4.6% of NSCLCs demonstrated the expression imbalance, and the above estimates approached 170/224 (75.9%) and 170/185 (91.9%), respectively. The majority (176/185 (95.1%)) of common RET rearrangements were detectable if 5’/3’ expression imbalance assay utilized the threshold >/=1, however only 176/376 (46.8%) of tumors with this relaxed cut-off carried RET fusion (Supplementary Tables S3, S4; Figures S1).
…This study utilized combination of 5’/3’-end unbalanced expression assay with the deltaCt >/=3 and one multiplexed PCR reaction for eight the most common RET translocations as the screening procedure. It may be feasible to reduce the threshold for the deltaCt down to >/=2 or even to >/=1, as it will result only in modest increase of the proportion of samples undergoing further analysis (Supplementary Table S3). Comprehensive PCR-based testing of these samples is non-expensive, as it requires only 3 additional multiplexed reactions capable of recognizing 29 RET translocation variants. However, the relaxation of the deltaCt threshold will result in dramatic increase of the number of samples, which are negative for 8 common and 29 rare translocations upon PCR analysis (Supplementary Tables S3, S4; Figure S1). It is questionable, whether the PCR-negative samples with the deltaCt threshold above 1 but below 3 deserve to be subjected to NGS. The choice of the optimal cut-offs and the depths of the analysis of NSCLCs with border-line deltaCt values may depend on the local experience and the available resources.
Comment: The authors should comply with the HUGO Gene Nomenclature Committee (HGNC) recommendations for the designation of gene fusions (PMID: 34615987). Thus, due to shortcomings of using in hyphen (-) or a forward slash (/) to describe gene fusions, HGNC has worked with the scientific community to determine a new, instantly recognizable and unique separator-a double colon (::) to be used in the description of fusion genes, and advocates its usage in all databases and articles describing gene fusions.
Response: This is done.

Reviewer 2 Report
The manuscript submitted by Tiurin and colleagues reported the use of a RT-PCR-based 5’/3’ end expression test and a variant specific PCR for rapid and cost-efficient detection of RET rearrangements in NSCLC.
Given the key importance of testing for druggable alterations in NSCLC, the presented study could significantly contribute to define effective detection methods for RET fusions.
Nevertheless the study design should be improved and the manuscript is confusing and inaccurate in some parts, it needs a robust editing and rephrasing in order to better report the study design, the results and mainly the discussion of the obtained data.
Examples of key points that need to be better described/discussed:
My understand is that the NGS analysis to compare with the presented method data was conducted after the RT-PCR-based 5’/3’ end expression tests were done: it would more accurate to include also a significant number of cases with a known/confirmed RET fusions that are then analysed for the RT-PCR-based 5’/3’ end expression.
The pemetrexed topic is not even mentioned in the abstract, and it is difficult for the reader to understand the meaning of the relationship between this drug and the presence of RET fusions in the context of this study aim.
The >10% of inadequate cDNA obtained (line 79) must be adequately discussed because suche a high percentage is a big limit in the proposed routine use of this diagnostic approach
(Lines 99-114) How had the threshold >/=3 been chosen? As a quite relevant number of cases (23) “showed a trend a trend towards the 5’/3’-end RET expression imbalance, although the deltaCt was below a predefined threshold”, could an intermediate deltaCt value be proposed to define a grey zone that needs to be investigated by additional methods?
(Lines 109-123) This part is not sufficiently described in the material and methods.
(Lines 117-123) What is the relevance of these data in respect of the study aim?
(Lines 147-150) This key part about the sensitivity of the test mjust be better discussed in order to support the usefulness of this approach.
English needs major revisions.
Author Response
Comment: My understand is that the NGS analysis to compare with the presented method data was conducted after the RT-PCR-based 5’/3’ end expression tests were done: it would more accurate to include also a significant number of cases with a known/confirmed RET fusions that are then analysed for the RT-PCR-based 5’/3’ end expression.
Response: We now comment on this issue in the Results section:
This study included 4849 tumors, which were analyzed both by 5’/3’-end expression imbalance test and by variant-specific PCR for eight common RET translocations. These data provided an opportunity for rough evaluation of the specificity and sensitivity of the former assay. It was revealed that the performance of 5’/3’-end expression test was significantly influenced by the threshold chosen. For example, only 1.3% of the total number of the analyzed carcinomas demonstrated the 5’/3’-end imbalance with deltaCt >/=3; 152/169 (89.9%) tumors meeting this cut-off carried common RET rearrangements, however only 152/185 (82.2%) NSCLCs with RET fusions had deltaCt satisfying this cut-off. When the threshold was brought down to >/=2, 4.6% of NSCLCs demonstrated the expression imbalance, and the above estimates approached 170/224 (75.9%) and 170/185 (91.9%), respectively. The majority (176/185 (95.1%)) of common RET rearrangements were detectable if 5’/3’ expression imbalance assay utilized the threshold >/=1, however only 176/376 (46.8%) of tumors with this relaxed cut-off carried RET fusion (Supplementary Tables S3, S4; Figures S1).
Comment: The pemetrexed topic is not even mentioned in the abstract, and it is difficult for the reader to understand the meaning of the relationship between this drug and the presence of RET fusions in the context of this study aim.
Response: We have rephrased the appropriate statement in the Abstract: ”RET-rearranged tumors obtained from females, but not males, had decreased level of expression of thymidylate synthase (p < 0.00001), which is a known predictive marker for efficacy of pemetrexed.”
Comment: The >10% of inadequate cDNA obtained (line 79) must be adequately discussed because such a high percentage is a big limit in the proposed routine use of this diagnostic approach (Lines 99-114).
Response: We have inserted a paragraph in the Discussion section addressing this issue:
This study failed to obtain cDNA of sufficient quality in 1657/16106 (10.3%) NSCLCs. It is essential to emphasize that NSCLC specimens were collected from various hospitals, therefore some inconsistencies in the processing and formalin fixation of tumor samples could have contributed to this relatively high failure rate. Surprisingly, most of the published large-scale studies did not report the proportion of samples unsuitable for RET analysis [10, 13, 14, 18, 22, 25-28]. The available data indicate that the quality of NSCLC tissues is essential for all RET testing procedures. Tsuta et al. [21] considered 1927 samples obtained in a single cancer center and observed a FISH failure in 53 (2.8%) tumors. Similarly, only 2/121 (1.7%) were unsuitable for FISH testing in the study of Baker et al. [11]. Failure rates are apparently higher for RNA-based techniques. Reguart et al. [29] reported insufficient cDNA quality in 7/108 (6.5%) samples analyzed. The proportion of NSCLCs with degraded RNA approached 6/30 (20%) in the study of Piton et al. [30]. To our knowledge, there were no investigations, which rigorously compared failure rates across various RET testing platforms.
Comment: How had the threshold >/=3 been chosen? As a quite relevant number of cases (23) “showed a trend a trend towards the 5’/3’-end RET expression imbalance, although the deltaCt was below a predefined threshold”, could an intermediate deltaCt value be proposed to define a grey zone that needs to be investigated by additional methods?
Response: We now comment on this issue in Results and Discussion sections, with the mention of newly added Table S3, Table S4 and Figure S1:
This study included 4849 tumors, which were analyzed both by 5’/3’-end expression imbalance test and by variant-specific PCR for eight common RET translocations. These data provided an opportunity for rough evaluation of the specificity and sensitivity of the former assay. It was revealed that the performance of 5’/3’-end expression test was significantly influenced by the threshold chosen. For example, only 1.3% of the total number of the analyzed carcinomas demonstrated the 5’/3’-end imbalance with deltaCt >/=3; 152/169 (89.9%) tumors meeting this cut-off carried common RET rearrangements, however only 152/185 (82.2%) NSCLCs with RET fusions had deltaCt satisfying this cut-off. When the threshold was brought down to >/=2, 4.6% of NSCLCs demonstrated the expression imbalance, and the above estimates approached 170/224 (75.9%) and 170/185 (91.9%), respectively. The majority (176/185 (95.1%)) of common RET rearrangements were detectable if 5’/3’ expression imbalance assay utilized the threshold >/=1, however only 176/376 (46.8%) of tumors with this relaxed cut-off carried RET fusion (Supplementary Tables S3, S4; Figures S1).
… This study utilized combination of 5’/3’-end unbalanced expression assay with the deltaCt >/=3 and one multiplexed PCR reaction for eight the most common RET translocations as the screening procedure. It may be feasible to reduce the threshold for the deltaCt down to >/=2 or even to >/=1, as it will result only in modest increase of the proportion of samples undergoing further analysis (Supplementary Table S3). Comprehensive PCR-based testing of these samples is non-expensive, as it requires only 3 additional multiplexed reactions capable of recognizing 29 RET translocation variants. However, the relaxation of the deltaCt threshold will result in dramatic increase of the number of samples, which are negative for 8 common and 29 rare translocations upon PCR analysis (Supplementary Tables S3, S4; Figure S1). It is questionable, whether the PCR-negative samples with the deltaCt threshold above 1 but below 3 deserve to be subjected to NGS. The choice of the optimal cut-offs and the depths of the analysis of NSCLCs with border-line deltaCt values may depend on the local experience and the available resources.
Comment: (Lines 109-123) This part is not sufficiently described in the material and methods.
Response: We have provided relevant details in the revised version of the paper:
… EGFR analysis was performed as described in [33]. The procedure for ALK and ROS1 testing is given in detail in [20]. The detection of MET exon 14 skipping mutations was carried out in accordance with the study [34]. KRAS and BRAF mutation analysis is outlined in [35].
Comment: (Lines 117-123) What is the relevance of these data in respect of the study aim?
Response: We have incorporated an introductory sentence: “Clinical characteristics of patients with RET translocations were considered.”
Comment: (Lines 147-150) This key part about the sensitivity of the test must be better discussed in order to support the usefulness of this approach.
Response: In addition to the describing the distribution of RET rearrangements in samples with different 5’/3’-end expression ratios (see above), we have incorporated the following paragraph in Discussion section:
Systematic evaluation of the actual sensitivity of the proposed procedure requires its comparison towards the “gold standard”, i.e., RNA-based NGS. It is particularly relevant to perform these validation experiments for NSCLCs, which have increased probability of carrying a kinase-activating translocation, e.g., tumors diagnosed in young and/or female non-smokers, but negative for known driver mutations detected by standard methods [6, 13, 21, 23, 24]. By the time of the manuscript preparation, RNA-based NGS was accomplished for 87 NSCLCs, which satisfied the above clinical criteria and had no evidence for ALK, ROS1, RET or NTRK1-3 gene rearrangements according to 5’/3’-end expression test and variant-specific PCR. Strikingly, NGS analysis revealed that none of these carcinomas carried translocation in the above genes [24]. Despite these encouraging data, further efforts in avoiding even the minimal risk of false-negative negative results may be justified, especially given that the magnitude of benefit from targeted therapy in patients with RET-rearranged NSCLC is indeed dramatic. …
The revised version of the manuscript has been edited by Dr. Priscilla S. Amankwah, who is a native English speaker proficient in scientific biomedical writing.

Round 2
Reviewer 1 Report
The authors have successfully addressed the concerns from the previous version.
Author Response
Thank you very much!!!
Reviewer 2 Report
The authors definitely improved the quality of the manuscript and satisfactorily answered to all comments and questions, therefore the manuscript is now suitable for publication.
Minor comments:
Tables S1, S2, S3 should be included in the main text as they significantly help the reader to go through data and results.
Line 142: “….152/185 (82.2%) NSCLCs with RET fusions had deltaCt satisfying this cut-off”: why the number of the proved RET fusions is 152 instead of 163? Even though 11 cases (3+8) were found positive for RET rearrangement by additional methods, they are positive and contribute to increase the significance of the 5’/3’ imbalance deltaCt cut-off.
English is fine, only minor editing needed
Author Response
Thank you very much for encouraging evaluation! Please find below the responses:
Comment: Tables S1, S2, S3 should be included in the main text as they significantly help the reader to go through data and results.
Response: Thank you very much for this suggestion, this is done. Of course, we have modified the numbers of all remaining Tables accordingly.
Comment: Line 142: “….152/185 (82.2%) NSCLCs with RET fusions had deltaCt satisfying this cut-off”: why the number of the proved RET fusions is 152 instead of 163? Even though 11 cases (3+8) were found positive for RET rearrangement by additional methods, they are positive and contribute to increase the significance of the 5’/3’ imbalance deltaCt cut-off.
Response: We are now providing these estimates both for 8 common rearrangements and for fusions identified by additional methods:
… For example, only 1.3% of the total number of the analyzed carcinomas demonstrated the 5’/3’-end imbalance with deltaCt >/=3; 152/169 (89.9%) tumors meeting this cut-off carried common RET rearrangements, however only 152/185 (82.2%) NSCLCs with RET fusions had deltaCt satisfying this cut-off (Tables 1, 3). When we considered not only rearrangements detected by variant-specific PCR for eight common fusions, but also included RET alterations identified by other tests, the latter value did not change significantly and approached 163/197 (82.7%) (Table 1). ….